# Advanced Pressure Compensation in High Accuracy NDIR Sensors for Environmental Studies

**DOI:** 10.3390/s23052872

**Published:** 2023-03-06

**Authors:** Bakhram Gaynullin, Christine Hummelgård, Claes Mattsson, Göran Thungström, Henrik Rödjegård

**Affiliations:** 1Senseair AB, 824 71 Delsbo, Sweden; 2Department of Engineering, Mathematics and Science Education, Mid Sweden University, 851 70 Sundsvall, Sweden

**Keywords:** NDIR, sensor, pressure, compensation, carbon dioxide

## Abstract

Measurements of atmospheric gas concentrations using of NDIR gas sensors requires compensation of ambient pressure variations to achieve reliable result. The extensively used general correction method is based on collecting data for varying pressures for a single reference concentration. This one-dimensional compensation approach is valid for measurements carried out in gas concentrations close to reference concentration but will introduce significant errors for concentrations further away from the calibration point. For applications, requiring high accuracy, collecting, and storing calibration data at several reference concentrations can reduce the error. However, this method will cause higher demands on memory capacity and computational power, which is problematic for cost sensitive applications. We present here an advanced, but practical, algorithm for compensation of environmental pressure variations for relatively low-cost/high resolution NDIR systems. The algorithm consists of a two-dimensional compensation procedure, which widens the valid pressure and concentrations range but with a minimal need to store calibration data, compared to the general one-dimensional compensation method based on a single reference concentration. The implementation of the presented two-dimensional algorithm was verified at two independent concentrations. The results show a reduction in the compensation error from 5.1% and 7.3%, for the one-dimensional method, to −0.02% and 0.83% for the two-dimensional algorithm. In addition, the presented two-dimensional algorithm only requires calibration in four reference gases and the storing of four sets of polynomial coefficients used for calculations.

## 1. Introduction

Continuous development and improvement of non-dispersive infrared (NDIR) gas sensors has allowed them to reach very high resolution, ppm and sub-ppm, in relatively low-cost designs. With high mechanical accuracy of the optical system, temperature-controlled optics, highly stable electronics, and optimized mirror surfaces, a detection level as low as 0.007 ppm can be achieved [1].

The theoretical principle of NDIR sensors, as described by the Lambert–Beer law [2], is the dependence between the transmittance of light and the number of gas molecules capable of absorbing the light. The total number of molecules involved in the light absorption depends on the volumetric concentration and the ambient pressure in sensed volume. This means that even small changes in the ambient pressure can easily result in measurement errors exceeding the required accuracy specified for the application.

Hence, for demanding applications, such as environmental and atmospheric research [3], there is a need for a reliable solution that can compensate for the influence of the ambient pressure variations. Significant improvements in the measurement performance can be attained by implementation of an accurate pressure compensation procedure.

To our knowledge, only a few research studies have been published on the topic of the implementation of an accurate pressure compensation procedure in NDIR applications. However, in [4], Robert Frodl et.al presented a spectroscopic gas sensor system for measurement of carbon dioxide. They briefly discussed the compensation of atmospheric pressure variations utilizing a piezoresistive pressure sensor, but without presenting details of the method. In addition, pressure compensation in a NDIR sensor has also been studied in [5,6,7] without any details regarding the method of choice. 

References [8,9,10] discuss the pressure compensation approaches, which are usually applied during post-processing of measured results. These are mostly based on a simple linear regression recalculation method.

Some information about linear compensation coefficients can be found in manuals and application notes from NDIR sensor manufacturers [11,12].

One of the most advanced approaches, which consider the concentration dependence of the compensation parameters, can be found in [13]. However, this compensation algorithm uses theoretical parameters derived from spectral simulation. This also means that variations found in real sensor system are neglected in the compensation parameters.

In general, the most common pressure compensation algorithms, which are based on calculation of compensation parameters, are obtained at reference concentrations similar to the application. This approach, considered as a one-dimensional compensation method, has the ability to compensate for varying pressure close to the reference concentration. However, in reality, compensation parameters obtained at equal pressure values can vary significantly for different concentrations. This means, that the use of compensation parameters obtained in one specific reference concentration can lead to significant compensation errors for other concentrations. One way of handling the concentration dependence, is to determine the compensation parameters at many reference concentrations. Unfortunately, this will require a complicated calibration procedure, which involves a large number of reference gases. It will also require a large number of compensation parameters being stored to cover the whole concentration range in the intended application.

In this paper, we present an advance compensation algorithm, which offers a practical solution based on empirical results obtained in precisely controlled conditions. It allows for the implementation of a pressure compensation, which is individual and includes all possible uncertainties found in a specific sensor. The presented result shows that the algorithm is valid over a relatively broad concentration range, and sets minimum memory and computational requirements for the sensor system.

## 2. Motivation for Calibration-Based Compensation

As mentioned in the introduction, the transmittance of specific wavelength of light traveling through a volume of gas is described by the Lambert–Beer law [2,14] and shown in Figure 1:

For an ideal gas at pressure ***P*** and temperature ***T***, the mixing ratio ***q***, corresponding to the volumetric or fractional concentration, expressed in parts per million (ppm) equals [14]:
(1)$$q=\frac{kT}{P}\times \frac{1}{x\sigma_{\vartheta }}ln\left(\tau \right)$$where ***τ*** is the radiation transmittance, ***k*** is the Boltzmann constant, ***x*** optic path length. The spectral dependence is contained in the absorption cross section, ***σ_ϑ_*** (area/molecule).

As seen, in Equation (1) there is a direct dependence between concentration, gas pressure P, and transmittance. Two parameters in this equation contains a pressure-variation dependency, which have an impact on the concentration reported by the sensor:
The gas pressure ***P***, which affects the density of light absorbing molecules in the optical path;The absorption cross section ***σ_ϑ,_*** which represents the spectral absorptivity of a specific gas, summed from the absorption peaks of the gas molecules.

It may be perceived as fairly simple to use Equation (1) as a theoretical model for pressure compensation, if the sensor continuously measures the pressure and have access to stored data regarding the absorption cross section (available from [15]).

Unfortunately, in a practical implementation with a large number of individual sensors and the need of high accuracy, there are several other factors that complicate theoretical modelling.

The variation of the number of molecules in the optic path, due to variation in pressure, has a direct impact and is used for simple modelling. Application notes and recommendation found in [11,12], propose approaches for the compensation of the influence of pressure on the density of absorbing molecules. These are based on a simple linear dependence between variation of pressure and the sensors reported concentration. They can provide compensation for this type of impact within a limited concentration range close to reference concentration conditions. For the most common sensors and applications, with tolerances of ±30 ppm and ±3% of sensors reading [16], this can be satisfactory. However, tougher demands regarding accuracy in gas concentration measurements, necessitate development of improved methods which include all pressure-dependent aspects. This includes handling of the effect of the variation in absorption cross section ***σ_ϑ_***, caused by variation in pressure.

Figure 2 presents the simulation results of absorption cross sections, ***σ_ϑ_*** [15], for an environment with stable CO_2_ concentration and temperature, for two different values of pressure. As can be seen, there is a clear difference in the resultant absorption cross section due to the variation of pressure. According to Equation (1), a decrease in the absorption cross section results in an increase in the concentration reported by the sensor. This means that pressure variation in the sensing environment affects the optical transmission in two ways:
Quantitatively, as a change in the number of absorbing molecules in the measured volume (sensor cavity);Qualitatively, as changes in the molecules’ absorbing capacity.

Hence, even for ideal conditions theoretical modelling of the variations in absorptivity caused by the variations in pressure is complicated. In addition, a practical system implementation will result in physical uncertainties and variations which are also difficult to model theoretically and to compensate for. These include:
The variation in optic filter passbands, due to individual variations;The wide range of angle of incidence (AOI) for light beams crossing optic filters.

Figure 2 presents a transmission simulation for light with AOI = 0 degrees.

In a calibration-based compensation procedure, which is an empirical method, all deviations are taken into consideration by collecting calibration data in the reference conditions. For large-scale production of devices with limited computational and storage resources, the calibration approach utilized in the presented work, is the preferable method.

## 3. General One-Dimensional Pressure Compensation Method

### 3.1. Definitions

To present the general method for pressure compensation of a NDIR sensor, some definitions must be introduced:***q_meas_*** is the concentration value directly reported by the sensors without applied pressure compensation. Errors caused by pressure variations in the environment are included in ***q_meas_***;***q_P0_*** is the concentration value reported by sensor under the standard pressure conditions P_0_ = 1.013 Bar, all non-pressure-dependent measurement errors are herein considered negligible;***q_ref_*** is the reference concentration;***q_comp_*** is the concentration value calculated from ***q_meas_*** by the pressure compensation method.

In ideal conditions, ***q_P0_*** should equal to ***q_ref_***. In reality, the sensors could show deviations within specified tolerances. Hence, the value of ***q_comp_*** should be targeted to be equal to the sensor’s own reading in standard pressure condition ***q_P0._***

### 3.2. Derivation of Compensation Coefficients

Equation (2) presents the compensated concentration for measurements performed at a pressure ***P*** ≠ ***P*_0_**:(2)$$q_{comp}=\frac{q_{meas}}{K}$$
where ***K*** is the pressure compensation factor at a specific pressure, obtained through calibration.

The general pressure compensation method, which is widely used [11,12], requires the compensation factor value ***K*** to be obtained as a function within the working pressure range of the sensor. ***K*** is derived from a limited number of measurements done on certain pressure points to be obtained. Since ***K*** is a function, this allows the compensation factor for pressure points even between the actual measuring points.

Pressure calibration procedure is performed in a pressure variable environment with stable concentration and temperature conditions. Pressure calibration equipment provides a set of environments with resolution 100 mBar. On each pressure step ***q_meas_*** is obtained and the respective compensation factor is calculated:(3)$$K=\frac{q_{meas}}{q_{P0}}$$

The curve fitting for calibration data points give a polynomial equation for compensation factor function:(4)$$K=A{\left(P-P_{0}\right)}^{2}+B\left(P-P_{0}\right)+1$$

A second-order polynomial is motivated by the fact that a linear dependence gives an error in the linear fit (square sum) within 0.2 to 1.0%, whereas a second-order polynomic results in an error below 0.04%. Increasing the polynomial to third order only provides an additional accuracy of 0.01%. This could be considered as negligible and requires greater computational demands. ***A*** and ***B*** are single polynomial coefficients, which are stored in the sensor’s microprocessor memory.

Figure 3 presents a calibration performed in a CO_2_ environment equal to ***q_ref_*** = 400 ppm to obtain the pressure dependence of the compensation factor ***K***.

The data collected and calculated in the calibration points give the compensation factor ***K*** vs. delta pressure (***P – P*_0_**). This could be described by polynomial Equation (4), which from Figure 3 results in:(5)$$K=0.2923{\mathrm{Bar}}^{-2}{\left(P-P_{0}\right)}^{2}+1.3019{\mathrm{Bar}}^{-1}\left(P-P_{0}\right)+1$$

Table 1 contains the verification results for two arbitrary pressure values taken in the reference environment; here ***∆*** is deviation of compensated value ***q_comp_*** from ***q_P0_***.

Compensated concentration values obtained for pressure points, which are between calibration points give accurate results within a sensor’s specified tolerances [16].

### 3.3. Concentration Dependence of the Compensation Factor

The general pressure compensation method could be considered as a one-dimensional method, which allows for efficient and accurate compensation along pressure variation at one specific concentration.

Figure 4a shows an example of how the compensation factors for varying pressures, calculated using Equation (4), differ depending on the CO_2_ concentration. The magnification in Figure 4b clearly shows how the compensation factor for a specific pressure varies (red dots) with the concentration.

Figure 5 presents the values for the compensation factor as 3D surface in the investigated pressure and concentration range. Each reference concentration for a certain pressure level is matched with a set of ***K*** values (red dots).

The variation in the compensation factor at the same pressure level, will introduce errors in the compensated value unless ***q_meas_*** is close to the reference value where the factor was derived.

To improve the accuracy of the pressure compensation at any point in the pressure-concentration space, it is therefore important also to consider the concentration dependence.

Thus, to resolve the impact from concentration dependence requires interpolation in a second dimension.

## 4. Experimental

A calibration procedure always requires the use of initial data, which were collected against reference values. In the case of pressure calibration, this requires stable pressure and concentration conditions.

The one-dimensional compensation method uses initial data collected in one reference concentration near to the application range. The introduced two-dimensional algorithm requires at least two reference concentrations. The use of additional reference concentrations provides better performance but also requires more computational and memory capacity in sensor design.

The concentration range and the number of reference concentrations, used in the pressure calibration procedure, is determined by the field of application. Usually, in the calibration, the concentration range should be wider than the required application range. For environmental monitoring and indoor ventilation applications [17,18], a calibration range between 200 and 5000 ppm CO_2_ (under standard pressure ***P*_0_** = 1.013 Bar) is considered as sufficient since the normal working range is expected to be between 400 and 3000 ppm. 

For each reference concentration, the test system provides a stepped pressure variation inside the test volume within 0.5–1.1 Bar. This range corresponds to the pressures where human activity on earth is most common. Pressure ***P*_0_** = 1.013 Bar is the standard value at sea level. For experimental purposes and for future interpolation validation, the pressure variation is conducted in small steps (10–50 mBar depending on the specific range). When the pressure is closer to standard pressure, a smaller step is applied. 

The calibration system, presented in Figure 6, consists of a calibration chamber with pneumatic components, which supply reference gas and pressure stabilization at the respective pressure point during whole calibration procedure. The overall system is controlled by a LabVIEW-based software, which collects and logs calibration data for the subsequent calculation of the compensation parameters. A more detailed description of the system is presented in [14].

To evaluate the pressure compensation procedure, a calibration was performed based on six reference gases (200, 500, 800, 1600, 3000, and 5000 ppm CO_2_). Figure 7 presents the obtained data in four concentrations—200, 500, 1600, and 5000 ppm CO_2_, used for pressure calibration calculations. The data retrieved from reference concentration 800 and 3000 ppm was used for verification of the compensation procedures.

One pressure point from each of the reference concentrations 800 and 3000 ppm was selected and later used as a verification point. Table 2 presents the complete data for the verification points. In Table 2, the measured concentration by the sensor, ***q_meas_***, at the verification point is named ***Q__ver_***. 

## 5. Implementation and Verification of the Two-Dimensional Compensation Algorithm

As presented in Section 3.3, the pressure compensation factor has a concentration dependence. Thus, the pressure compensation algorithm should have the capability to derive the compensation factor at any point in the pressure–concentration space.

The developed pressure compensation algorithm operates using the concentration value ***q_meas_*** reported by NDIR sensor, the ambient pressure value P (obtained from either an internal pressure sensor or from external sources), and the individual calibration parameters derived in production calibration.

The calibration procedure provides data obtained in several reference environments with variable pressure. The compensation factor, as a set of functions ***K*** vs. (***P* − *P*_0_**), was calculated using Equation (3) for all reference concentrations. Figure 8 presents the dependences and their polynomial equations:

The polynomial coefficients in Equation (4) from the four references concentrations are summarized in Table 3:

Based on the verification parameters in Table 2, an implementation of the compensation algorithm could be evaluated and the compensation tolerance could be estimated.

The first interpolation step is performed along the pressure variation. The goal is to derive a compensation factor at each reference concentrations for the pressure ***P_ver*** = 0.72 Bar. Interpolative calculations also give the compensation factor between the pressure calibration points.

In the next step, the intermediate concentration ***q_n_norm_p_*** is introduced and calculated using previously obtained compensation factors:(6)$$q_{n\_norm\_P}=q_{P0}K_{n}$$

Here ***q_n_norm_p_*** is the normalized value of ***q_P0_*** at a pressure value ***P*** for the reference condition ***n***. Basically ***q_n_norm_p_*** is the concentration expected for the respective reference gas ***n*** at a pressure value ***P***. The space between ***q_n_norm_p_*** contains all possible ***q_meas_*** values at the same pressure ***P***.

Table 4 presents the compensation factors and respective expected concentrations ***q_n_norm_p_*** for the four reference dependences ***K__n_***.

Figure 9 presents the relationship between the compensation parameters and the concentration values.

In the second step of the algorithm, the dependence between ***K__n_*** vs. ***q_n_norm_p_***, plotted in Figure 10, is used for interpolation in the second dimension and calculation of the compensation factor for ***q_meas_ver_***:

The obtained dependence formula is:(7)$$K_{\_n}=3\times 10^{-8}{\mathrm{ppm}}^{-2}{\left(q_{n\_norm\_p}\right)}^{2}-0.0001{\mathrm{ppm}}^{-1}\left(q_{n\_norm\_p}\right)+0.6656$$

Since the reported values ***q_meas_*** and ***q_n_norm_p_*** belong to the same concentration space, the calculation of ***K__n_*** is performed through:(8)$$K_{\_n}=3\times 10^{-8}{\mathrm{ppm}}^{-2}{\left(q_{meas}\right)}^{2}-0.0001{\mathrm{ppm}}^{-1}\left(q_{meas}\right)+0.6656$$

Respectively, ***q_meas_ver_*** = 480 ppm in (8) gives the compensation factor ***K__ver_*** = 0.62. Now, by using (2), it is possible to calculate compensated value for ***q_meas_ver_*** = 480 ppm. Similar calculations were performed for an additional verification point ***q_meas_ver_*** = 1106 ppm CO_2_ and ***P__ver_*** = 0.55 Bar and plotted in Figure 11:

Table 5 summarizes all calculated parameters for both verification concentrations from Table 2.

## 6. Results and Discussion

In order to estimate the achieved improvement by adding a second interpolation dimension in the compensation algorithm, an error estimation was performed for two similar measurement conditions.

For HVAC (heating, ventilation, and air conditioning) or indoor air quality management applications, with a working range between 400 ppm and 3000 ppm, pressure calibration is normally performed at 1600 ppm. It is in the middle of the range and therefore a commonly occurring value.

The compensated concentration values ***q_comp,_*** was calculated for the two verification concentrations (800 ppm and 3000 ppm) using both the one-dimensional method (based on a compensation factor obtained at 1600 ppm) and the presented two-dimensional algorithm. The resulting values and errors relative to ***q_P0_*** are presented in Table 6.

The results presented in Table 6 clearly show a much higher accuracy for the two-dimensional pressure compensation algorithm compared to the general one-dimensional algorithm.

However, for the two-dimensional algorithm there is a significant difference in compensation error for the two verification points 480 ppm and 1096 ppm. The explanation can be found in Figure 10 and Figure 11 for the verification points respectively. The first point ***Q__ver_*** = 480 lays in range where the steps in the calibration concentrations are significantly shorter than for ***Q__ver_*** = 1096. Additionally, the dependence in transient compensation factors ***K__n_*** has a higher linearity in the lower concentration range.

To further improve the two-dimensional algorithm, a more thorough selection of reference environments, concentration step size, and their disposition within sensors’ working range may be performed.

## 7. Summary and Conclusions

In this work, we have presented a two-dimensional pressure compensation algorithm for NDIR sensors. For a typical indoor NDIR gas sensor application, with an accuracy of ±30 ppm and ±3% of sensor reading, the need for pressure compensations is limited. However, for demanding applications such as environmental and atmospheric research, pressure dependent concentration measurements can be a major limitation for this technique. The presented algorithm reduces the resulting compensation error caused by concentration dependence in the general one-dimensional compensation algorithm.

The general one-dimensional pressure compensation algorithm was initially investigated. The compensation parameters obtained in reference conditions, including all measurement uncertainties, were derived from the measured values. At a specific concentration, the compensated concentration, ***q_comp_*** shows deviation of less than 1% relative to the reported sensors concentration under standard pressure over the whole calibration pressure range.

The one-dimensional pressure compensation algorithm is widely used today. However, direct application of compensation parameters obtained in one reference concentration may lead to significant errors if these are used for compensation of concentrations away from this reference environment.

In order to solve the problem of concentration dependence, a second dimension was introduced.

From the physical point of view, a two-dimensional compensation makes sense since the molecule absorption cross section is pressure-dependent. A one-dimensional compensation can only handle the pressure effects from the ideal gas law and the absorption cross section variation at one specific concentration.

The functionality and accuracy of the presented two-dimensional algorithm was verified and estimated by calculating the compensation error for two independent reference concentrations. The resulting accuracy shows a compensation error of −0.02% and 0.83% for the reference values 800 ppm (at 0.72 Bar) and 3000 ppm (0.55 Bar), respectively. This could be compared to compensation errors of 5.1% and −7.3% achieved by the one-dimensional method in similar conditions. The accuracy of the one-dimensional method can be improved by obtaining calibrations parameters in many additional reference gas environments with high resolution. However, a calibration procedure involving a large number of reference gases (e.g., 20 concentrations), evaluated at several pressure points (e.g., approx. 8–10), would result in a need to store up to 200 compensation parameters. 

In contrast, the two-dimensional compensation algorithm presented in this work only requires calibration in four reference gases and the storage of four sets of polynomial coefficients used for calculations. Therefore, the implementation of a second dimension leads to a negligible increase in memory and computation requirements and requires a less extensive calibration procedure.

## Figures and Tables

**Figure 1 sensors-23-02872-f001:**
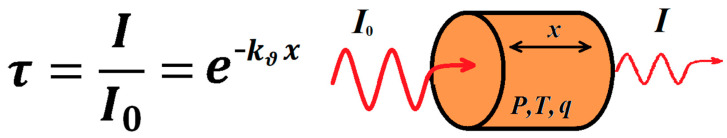
Lambert–Beer law’s graphic representation, ***k_ϑ_*** is the absorption coefficient. ***P***, ***T***, and ***q*** respectively pressure, temperature, and concentration in sensing volume.

**Figure 2 sensors-23-02872-f002:**
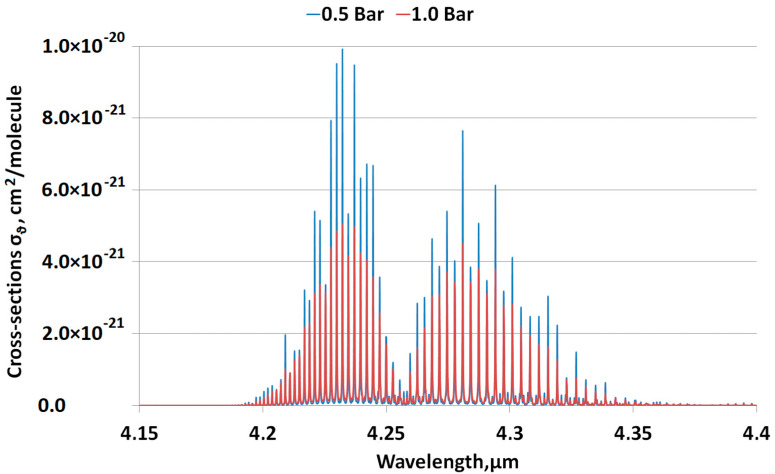
Simulation of absorption cross section ***σ_ϑ_*** for two pressure levels 0.5 and 1.0 Bar at a concentration of 400 ppm CO_2_ and a temperature of 296 K. A variation in ambient pressure leads to a variation in absorption capacity.

**Figure 3 sensors-23-02872-f003:**
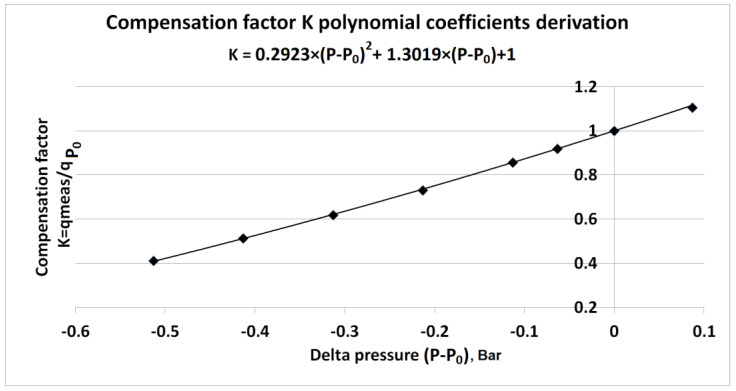
Pressure calibration procedure for deriving compensation factor function ***K*** with respective polynomial coefficients ***A*** = 0.2923 Bar ^−2^ and ***B*** = 1.3019 Bar ^−1^.

**Figure 4 sensors-23-02872-f004:**
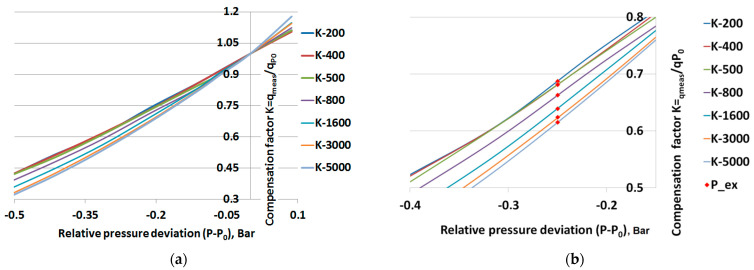
(**a**) Compensation factor ***K*** as function of relative pressure for different reference concentrations (200, 400, 500, 800, 1600, 3000, and 5000 ppm), (**b**) magnification of the relative pressure range −0.4 to −0.15 Bar for visually enhancing the concentration dependence of the compensation factor—red dots show difference in compensation factor values for different concentrations at the same pressure level ***P_ex***.

**Figure 5 sensors-23-02872-f005:**
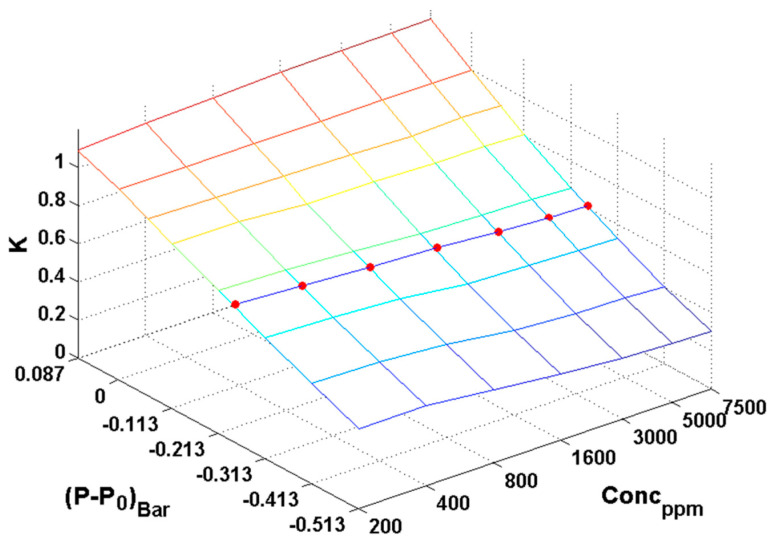
Space of compensation factor values in 3D volume concentration range vs. pressure range vs. compensation factor range. Red dots indicates ***K*** values for different reference concentrations on the same certain pressure level.

**Figure 6 sensors-23-02872-f006:**
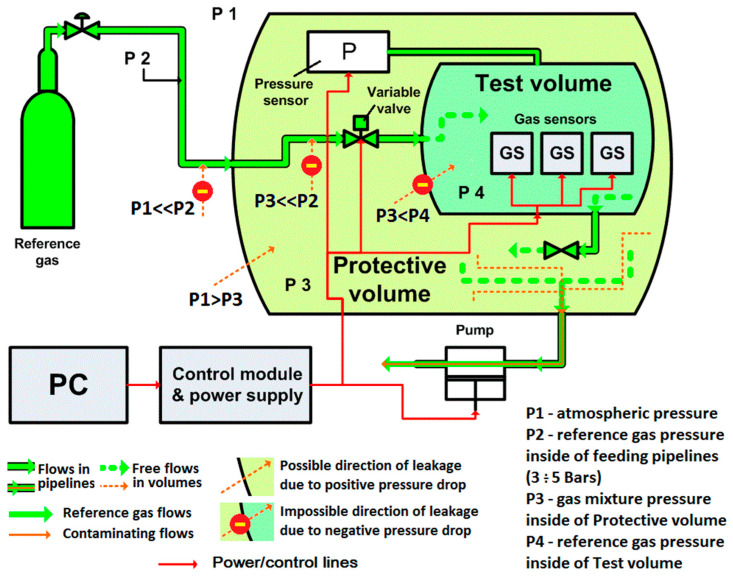
Schematic of the pressure calibration system. Protection against possible contamination from ambient concentration is based on a pressure drop. The pressure drop is developed by maintaining the pressure in the test volume higher than in the protection volume.

**Figure 7 sensors-23-02872-f007:**
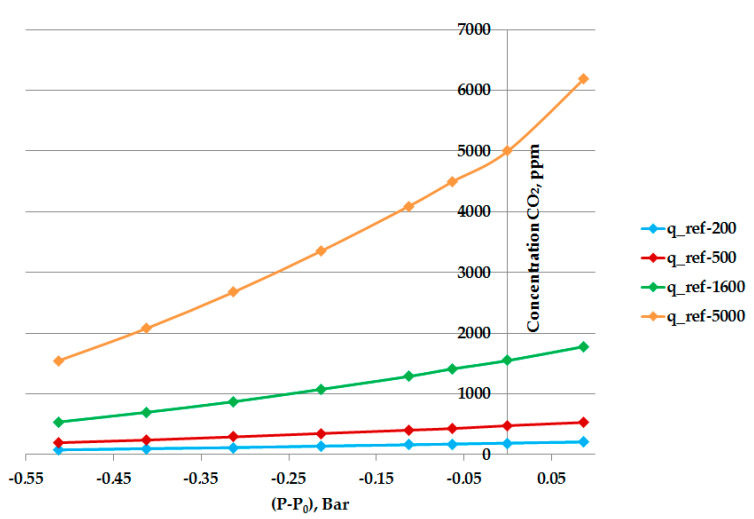
The pressure calibration data for reference concentrations 200, 500, 1600, and 5000 ppm CO_2_.

**Figure 8 sensors-23-02872-f008:**
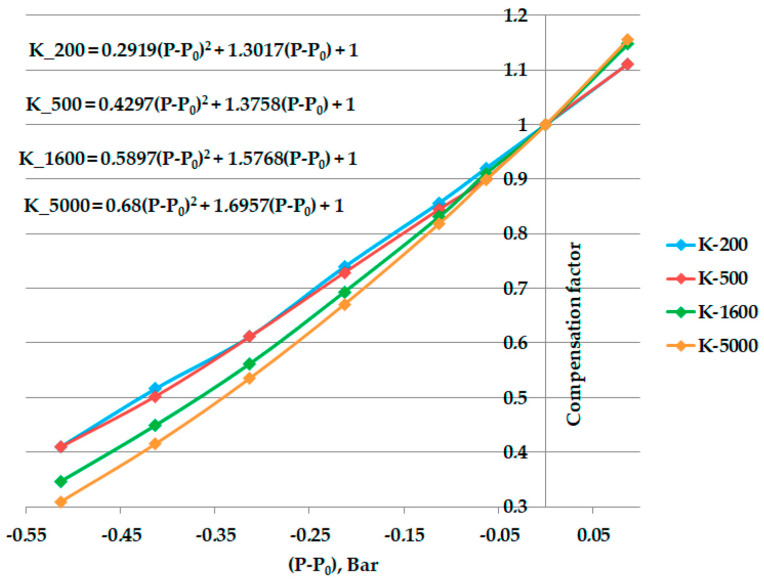
Compensation factor calibration dependences as a polynomial functions.

**Figure 9 sensors-23-02872-f009:**
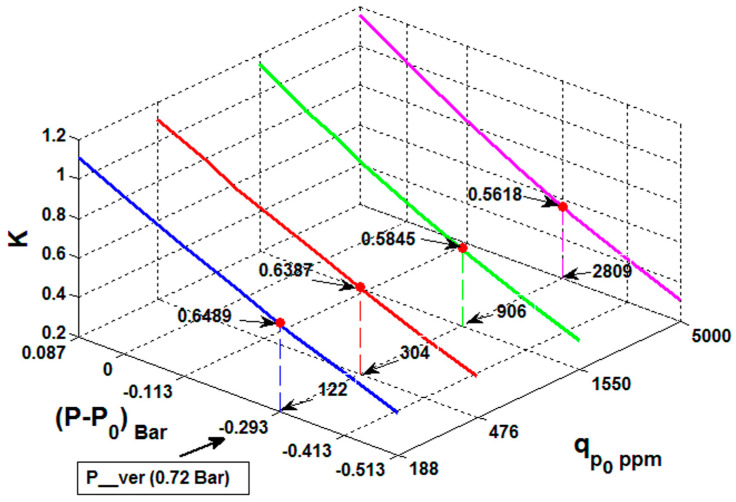
Overall relation between concentration and the compensation parameters. Red dots represent the compensation factor derived for the reference concentrations at pressure 0.72 Bar (equal to Table 4).

**Figure 10 sensors-23-02872-f010:**
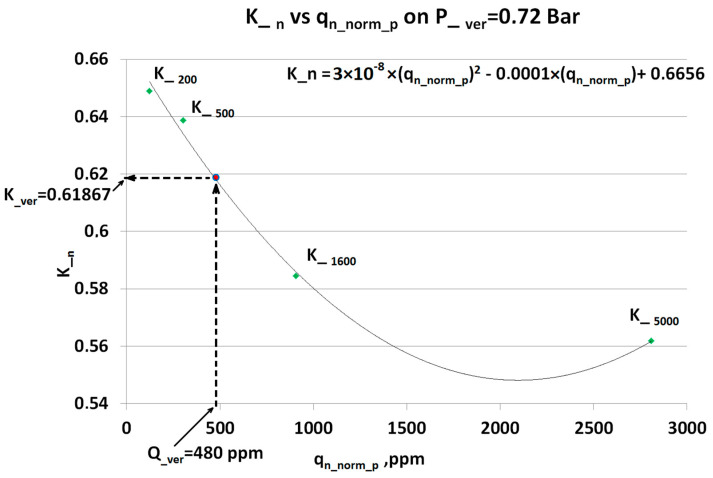
The dependence ***K__n_*** vs. ***q_n_norm_p_*** contains a set of all possible compensation factors within calibrated range at the specific pressure (here ***P__ver_*** = 0.72 Bar) where ***q_meas_*** was obtained. The ***K__n_*** dependence equation gives the value of the compensation (here ***K__ver_***) matching the respective reported ***q_meas_*** (here ***Q__ver_***).

**Figure 11 sensors-23-02872-f011:**
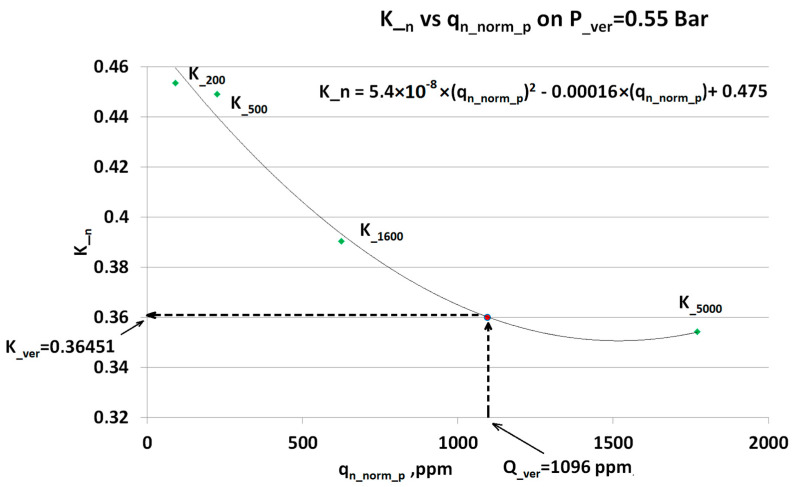
The dependence ***K__n_*** vs. ***q_n_norm_p_*** contains a set of all possible compensation factor within a calibrated range at the specific pressure (here ***P__ver_*** = 0.55 Bar) where ***q_meas_*** was obtained. The ***K__n_*** dependence-equation gives the value of the compensation (here ***K__ver_***) matching the respective reported ***q_meas_*** (here ***Q__ver_***).

**Table 1 sensors-23-02872-t001:** Verification of the compensation algorithm performed on two arbitrary pressure points.

P [Bar]	(P − P_0_) [Bar]	q_P0_ [ppm]	q_meas_ [ppm]	q_comp_ [ppm]	∆ [%]
0.8499	−0.1631	382	303	380.76	−0.33
1.0499	0.0369	382	397	379.42	−0.68

**Table 2 sensors-23-02872-t002:** Verification parameters.

P__ver_ [Bar]	(P − P_0_) [Bar]	q_ref_ [ppm]	q_P0_ [ppm]	Q__ver_ (q_meas_) [ppm]
0.72	−0.293	800	776	480
0.55	−0.633	3000	2982	1096

**Table 3 sensors-23-02872-t003:** Polynomial coefficients in Equation (4), for calculation of compensation factor for the four reference concentrations.

K	A, [Bar ^−2^]	B, [Bar ^−1^]
* **K_200** *	0.2919	1.3017
* **K_500** *	0.4297	1.3758
* **K_1600** *	0.5897	1.5768
* **K_5000** *	0.68	1.6957

**Table 4 sensors-23-02872-t004:** Compensated factors for P_ver on all calibration dependences.

P_ver [Bar]	K_200	K_500	K_1600	K_5000
0.72	0.65	0.64	0.58	0.56
***q_n_norm_p_***, ppm	122	304	906	2809

**Table 5 sensors-23-02872-t005:** Calculated parameters for selected verification points.

q_P0_, [ppm]	P_ver [Bar]	q_meas_ (Q__ver_) [ppm]	K__ver_	q_comp_ [ppm]	∆ [%]
776	0.72	480	0.62	775.86	−0.02
2982	0.55	1096	0.36	3006	0.83

**Table 6 sensors-23-02872-t006:** Compensated concentrations compared to true values for different concentrations ≠ pressure calibration standard.

	q_ref_ = 800 ppm	q_ref_ = 3000 ppm
** *q_meas_* **	480 ppm (***P*** = 0.72 Bar)	1096 ppm (***P*** = 0.55 Bar)
** *q_P_* _0_ **	776 ppm	2982 ppm
***K_1*** (one dimensional compensation factor obtained for 1600 ppm), Figure 3	0.59	0.40
***q_comp_*** (one--dimensional) according to Equation (2)	815.5 ppm	2765.2 ppm
**One-dim. compensation error, %**	5.1%	−7.3%
***q_comp_*** (two-dimensional), from Table 5	775.86 ppm	3006 ppm
**Two-dim. compensation error, %**	−0.02%	0.83%

## Data Availability

The data presented in this study are available on request from the corresponding author.

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
