# Peer review of "Advanced Pressure Compensation in High Accuracy NDIR Sensors for Environmental Studies"

_sensors, 2023, doi:10.3390/s23052872_

Round 1
Reviewer 1 Report
The work is devoted to the development of an efficient algorithm for atmospheric pressure compensation, which is necessary for measuring gas concentration with high resolution. The paper presents a two-dimensional algorithm that allows compensation in a wide range of pressures and concentrations using a minimum amount of data. The quality of the work is not inferior to the previous works of the authors. However, it would be interesting for readers of the Sensors journal to see at least one drawing of the experimental scheme or instrument used by the authors. For example, to reveal to the reader in more detail the phrase - "This work was focused on compensating for atmospheric pressure changes using a piezoresistive pressure sensor." The manuscript is difficult to read, especially the Introduction and Motivation. Often one has to look for additional information in the cited or other literature. The manuscript may be published after a major revision.
Additional notes:
l.12-17. “This compensation give acceptable results for measurements done in environments close to reference concentration, but introduce significant errors for concentrations further away from the calibration point. For cases where high accuracy is important this problem can be solved by collection and storage of calibration data in several reference concentrations. For cost sensitive applications this is not normally feasible due to high demands on computing and memory capacities as well as an expensive calibration procedure.” - General phrases that are more appropriate to write in the introduction.
l.27-28. “The latest advancements in Non-Dispersive Infrared (NDIR) gas sensors allow for very high resolution, ppm and sub-ppm, in relatively low-cost designs [1].” - Why not add here a few explanations and conclusions from article [1] to make it more convenient for the reader.
l.35-36. A phrase that doesn't explain anything. Specify which demanding or common applications are commonly used.
Eq.1. I couldn't open Ref.[13] and connect this equation with the Lambert-Beer law presented by the authors, for example, in the article [1]. An explanation needs to be added. "x" is probably the optical path (explain).
l.83-84. “However growing demand for improved accuracy in gas concentration measurements requires new methods.” - Growing demand again. What is typical NDIR accuracy, how much does pressure compensation can improve accuracy?
Table 1. Delta needs to be defined.
l.163. Perhaps it should be added here that the two-dimensional method will now be considered. And explain here or in the introduction the reason for choosing the 2D method. Since it is not entirely clear that NDIR automatically changes the concentration or are these separate experiments? That is, an explicit description of a typical NDIR sensor is required.
Figure 3 is desirable to make in the same or closer scales, since it is difficult to compare panels a and b.
l.271. 3E-08 in fig. 8 is not the same as 3E^(-08) in formulas 7 and 8.
Summary - two-dimensional compensation algorithm is described clearly and in detail. However, it is not clear how it is included in the NDIR, if everything is automated, then this is really a low-cost device.
Author Response
Reviewer report 1
Comments and Suggestions for Authors
The work is devoted to the development of an efficient algorithm for atmospheric pressure compensation, which is necessary for measuring gas concentration with high resolution. The paper presents a two-dimensional algorithm that allows compensation in a wide range of pressures and concentrations using a minimum amount of data. The quality of the work is not inferior to the previous works of the authors. However, it would be interesting for readers of the Sensors journal to see at least one drawing of the experimental scheme or instrument used by the authors. For example, to reveal to the reader in more detail the phrase - "This work was focused on compensating for atmospheric pressure changes using a piezoresistive pressure sensor." The manuscript is difficult to read, especially the Introduction and Motivation. Often one has to look for additional information in the cited or other literature. The manuscript may be published after a major revision.
Answer to reviewer 1
Thank you for the comments and useful suggestions on how to improve and clarify the motivation, aim, and results of this work. We have tried to answer and handle your comments and suggestions one-by-one below. Changes in the revised paper has been highlighted using track-changes. Major changes have been highlighted with yellow markings.
Suggestion: “However, it would be interesting for readers of the Sensors journal to see at least one drawing of the experimental scheme or instrument used by the authors. For example, to reveal to the reader in more detail the phrase - "This work was focused on compensating for atmospheric pressure changes using a piezoresistive pressure sensor."
Authors response: There is always a balance between including and omitting details from previously published results. Throughout the article, we have tried to focus on the development of the presented algorithm and its results. Thus, we have also deliberately chosen to refer to previous publications instead of presenting background information again where we have judged that it is not needed for the understanding of the article. However, in section 4 – Experimental, additional information has been added that provides additional information about which physical parameters that the test system controls. An additional figure which presents the pressure calibration system has been added to chapter 4. In the introduction, related research studies which, the authors are aware of, are mentioned. The phrase “This work was focused on compensating for atmospheric pressure changes using a piezoresistive pressure sensor." relates to the aim of reference [5] and does not relate to our work.
Suggestion: “The manuscript is difficult to read, especially the Introduction and Motivation. Often one has to look for additional information in the cited or other literature.
Authors response: We can understand that the introduction and motivation of the work needs improvement. Additional information has been added to improve this in the introduction. The revised paper has also been sent to professional native English grammar and language check to improve readability. The presented study is a based on previous published studies. It covers a rather broad technology area that is challenging to describe in one and the same article without losing focus on the current research question. Therefore, as described earlier, we have chosen to refer to previous works.
Additional notes:
l.12-17. “This compensation give acceptable results for measurements done in environments close to reference concentration, but introduce significant errors for concentrations further away from the calibration point. For cases where high accuracy is important this problem can be solved by collection and storage of calibration data in several reference concentrations. For cost sensitive applications this is not normally feasible due to high demands on computing and memory capacities as well as an expensive calibration procedure.” - General phrases that are more appropriate to write in the introduction.
Authors response: The abstract of the paper has somewhat been revised to more clearly follow the structure of (background, method and result). The purpose of the phrase is to introduce the problem statement in the abstract. Based on the reviewer comments, we have however also realized that this part also needed improvement in the introduction/motivation. Some more info with respective references about current tolerances and applications as well as about potential applications where more tight tolerances are required were added.
l.27-28. “The latest advancements in Non-Dispersive Infrared (NDIR) gas sensors allow for very high resolution, ppm and sub-ppm, in relatively low-cost designs [1].” - Why not add here a few explanations and conclusions from article [1] to make it more convenient for the reader.
Authors response: Examples of the latest advancements has been added to the introduction.
l.35-36. A phrase that doesn't explain anything. Specify which demanding or common applications are commonly used.
Authors response: Examples of demanding applications, where pressure variations are needed has been added.
Eq.1. I couldn't open Ref.[13] and connect this equation with the Lambert-Beer law presented by the authors, for example, in the article [1]. An explanation needs to be added. "x" is probably the optical path (explain).
Authors response: The variable x is the optical path length. This has now been clarified in connection to the equation. In addition, the Lambert-Beer law has been described. An additional figure presenting the principal of the Lambert-Beer law has been added.
l.83-84. “However growing demand for improved accuracy in gas concentration measurements requires new methods.” - Growing demand again. What is typical NDIR accuracy, how much does pressure compensation can improve accuracy?
Authors response: The accuracy of a typical low-cost NDIR sensor is about 30ppm and ±3% of sensors reading. This information has been added. As shown by the results, a pressure compensation based on a single reference concentration can lead to errors which are higher than the specified accuracy also quite close to the reference concentration.
Table 1. Delta needs to be defined.
Authors response: Additional explanation to the factor Delta in table 1 has been added.
l.163. Perhaps it should be added here that the two-dimensional method will now be considered. And explain here or in the introduction the reason for choosing the 2D method. Since it is not entirely clear that NDIR automatically changes the concentration or are these separate experiments? That is, an explicit description of a typical NDIR sensor is required.
Authors response: Unfortunately, we have some trouble to follow the reviewer thought with this comment. Section 3.3 describes the issues with the general one-dimensional algorithm and how to resolve the impact from the concentration dependence. It does not discussion the two-dimensional algorithm.
Figure 3 is desirable to make in the same or closer scales, since it is difficult to compare panels a and b.
Authors response: The magnification of figure 3a in Figure 3b has been added for visually enhancing the concentration dependence of the compensation factor. The red dots show difference in compensation factor values for different concentrations at the same pressure level P_ex. There is therefore no need of comparing the plots in a and b. Plotting them on the same scale will be difficult
l.271. 3E-08 in fig. 8 is not the same as 3E^(-08) in formulas 7 and 8.
Authors response: We have realized that equation 7 and 8 had some errors in data representation and have now been changed.
Summary - two-dimensional compensation algorithm is described clearly and in detail. However, it is not clear how it is included in the NDIR sensor system, if everything is automated, then this is really a low-cost device.
Authors response: That this stage we are working on the development of the algorithm and evaluation of is validity has been done by post-processing. The presented algorithm results in in 4 equations for the compensation factors(K_200,K_500, K_1600 and K_5000), which will in the implementation requires the storage of 8 values. In comparison, by solving the pressure compensation by collection and storage of calibration data in several reference concentrations would quickly result in much higher storage need. E.g. already evaluating 10 reference concentrations at 10 pressures would result in the need of storing 100 values. Additional text has been added in the summary and conclusion which clarify this. In the conclusion, an estimation and comparison with other methods now added in conclusion
Reviewer 2 Report
The work is mainly for an efficient pressure algorithm for compensation of environment pressure variations, which is needed for measurements of gas concentrations with high resolution. The applied two-dimensional algorithm allows compensation in a wide pressure and concentration range using a minimum amount of data. The manuscript shows that the implementation of the two-dimensional algorithm reduces the compensation error from 5.1% and 7.3% to -0.02% and 0.83%, respectively, compared to compensation based on a single reference concentration. The first interpolation dimension handles uncertainties introduced by variations in the number of molecules due to pressure variation. Whereas the second interpolation dimension handles uncertainties introduced by variation of absorption cross-section, causing the same pressure variation. The manuscript algorithm is mainly for compensation for atmospheric pressure variations utilizing a piezoresistive pressure sensor.
The present form of the manuscript can be published without any corrections.
Author Response
Reviewer report 2
Comments and Suggestions for Authors
The work is mainly for an efficient pressure algorithm for compensation of environment pressure variations, which is needed for measurements of gas concentrations with high resolution. The applied two-dimensional algorithm allows compensation in a wide pressure and concentration range using a minimum amount of data. The manuscript shows that the implementation of the two-dimensional algorithm reduces the compensation error from 5.1% and 7.3% to -0.02% and 0.83%, respectively, compared to compensation based on a single reference concentration. The first interpolation dimension handles uncertainties introduced by variations in the number of molecules due to pressure variation. Whereas the second interpolation dimension handles uncertainties introduced by variation of absorption cross-section, causing the same pressure variation. The manuscript algorithm is mainly for compensation for atmospheric pressure variations utilizing a piezoresistive pressure sensor.
The present form of the manuscript can be published without any corrections.
Answer to reviewer 2
Based on complementary comments and our own review of the article, we have made a few changes to clarify certain parts and correct certain mistakes. The article has also been submitted for English grammar and language review. Changes in the revised paper has been highlighted using track-changes. Major changes have been highlighted with yellow markings.
Reviewer 3 Report
This paper presents a two dimensional method for pressure compensation in commercial NDIR sensors, and it has higher accuracy than the conventional 1-D model. This paper has some merits, but some changes are needed:
1) Overall language and logic need improving, it is hard to read.
2) More detailed background study instead of just referencing the relevant papers.
3) In motivation, add quantitative analysis to show the importance of pressure compensation. For instance, ‘the pressure can affect the measurement by XX%’ or something like that.
4) Where is the data comes from for Figure 2?
5) Based on the plot and authors’ calculations, the compensation accuracy decreases with higher concentrations, why is that?
6) Add more data points in Table 6 to show the validity of the model, for instance, same pressure difference concentrations, more concentrations, etc. Also, what is the highest possible compensation error?
Author Response
Reviewer report 3
Comments and Suggestions for Authors
This paper presents a two dimensional method for pressure compensation in commercial NDIR sensors, and it has higher accuracy than the conventional 1-D model. This paper has some merits, but some changes are needed:
Answer to reviewer 3
Suggestion: …” This paper has some merits, but some changes are needed:”
Authors response: We thank the reviewer for the suggestion and have tried to follow the input as much as possible. Changes in the revised paper has been highlighted using track-changes. Major changes have been highlighted with yellow markings.
Overall language and logic need improving, it is hard to read.
Authors response: The revised paper has also been sent to professional native English grammar and language check to improve readability. In addition, we have tried to improve the introduction and to clarify the background and aim of the work. We have also tried to enhance the text in some areas to improve the logical order. However, we believe that the outline of the chapters in the paper follows a rather logic order according to:
- Presents the background, related work and aim/motivation
- Motivation of the use of calibration in comparison to pure theoretical modeling of influence from pressure variation.
- Description of the one-dimensional pressure compensation, definitions and its short comings(concentration dependence in the compensation parameters)
- Presents the experimental procedure of the pressure calibration.
- Presents the implementation of the two-dimensional algorithm and verification for two independent reference concentration.
- Presentation of the compensation results for the verification point and comparison to the one-dimensional algorithm.
- Conclusion of the work and the result.
More detailed background study instead of just referencing the relevant papers.
Authors response: There is always a balance between including and omitting details from previously published results. Throughout the article, we have tried to focus on the development of the presented algorithm and its results. Thus, we have also deliberately chosen to refer to previous publications instead of presenting background information again where we have judged that it is not needed for the understanding of the article. However, in the revised paper we have added additional text that gives more information about the referenced works.
In motivation, add quantitative analysis to show the importance of pressure compensation. For instance, the pressure can affect the measurement by XX%’ or something like that.
Authors response: It is difficult to give such number, since the needed accuracy is dependent on the application. However, accuracy of a typical NDIR sensor used for indoor applications is about 30ppm and ±3% of sensors reading. This information has been added. As shown by the results, a pressure compensation based on a single reference concentration can lead to errors which are higher than the specified accuracy also quite close to the reference concentration.
Where is the data comes from for Figure 2?
Authors response: Figure 2 presents a calibration measurement performed in a CO2 environment equal to qref = 400 ppm to obtain the pressure dependence of the compensation factor K. Additional information has been added how the calibration was performed is added above equation 3.
Based on the plot and authors’ calculations, the compensation accuracy decreases with higher concentrations, why is that?
Authors response: The compensation accuracy does not decrease because of higher concentration. This effect is explained below table 9. To enhance the visibility of this effect, an additional figure(fig.9) has been added. The decrease depends on:
- Larger step between the calibration concentrations at higher concentration
- In the lower concentration area, the estimated dependence is more linear.
Add more data points in Table 6 to show the validity of the model, for instance, same pressure difference concentrations, more concentrations, etc. Also, what is the highest possible compensation error?
Authors response: In our verification we have selected two concentrations which lies rather far from each other in the whole range from 200 – 5000 ppm. Conducting a verification at additional points would of course be possible, but also require rather much additional work. It would also require additional tubes of reference gases. In the range from 200 – 5000 ppm, we have currently used 6 reference concentrations. 4 for the compensation algorithm and additional 2 for verification.
Reviewer 4 Report
The article is very interesting, although it has a few shortcomings that should be removed:
1. Reading the article, it is not clear what the authors' contribution is. Authors should therefore clearly state what exactly they have done. For example, by adding one or two paragraphs to the Introduction that explicitly list the authors' main contributions to the article (preferably in bullet points).
2. "The test system, described in more detail in [13], consisted of a hardware test bench connected to a LabVIEW-based software and calculation algorithm." (lines 209-210). In the reviewer's opinion, the article would be better and more interesting if it (even briefly) described the test environment, and not only referred readers to the authors' earlier publication.
3. "The implementation of second dimension leads to negligible increasing of memory and computation requirements" (lines 333-334). This claim, in the light of the published results, is unjustified - it is not based on either experimental research or any attempts at estimation. Since this is a relatively important issue and the authors have rightly drawn attention to it, they should develop this topic in their article. The more so that the article for Sensors should be at least 16 pages long, so the authors should add 3 pages anyway to meet the editorial requirements.
Author Response
Reviewer report 4
Comments and Suggestions for Authors
The article is very interesting, although it has a few shortcomings that should be removed:
Answer to reviewer 4
Authors response to suggestions and Comments: We thank the reviewer for the suggestion and comments and have tried to follow the input as much as possible. Changes in the revised paper has been highlighted using track-changes. Major changes have been highlighted with yellow markings.
Reading the article, it is not clear what the authors' contribution is. Authors should therefore clearly state what exactly they have done. For example, by adding one or two paragraphs to the Introduction that explicitly list the authors' main contributions to the article (preferably in bullet points).
Authors response: The authors of this paper have not been aware of that Sensors has a requirement for "Author Contribution list" in their publications. The revised paper has been updated with additional paragraphs in the end which clarifies the authors contribution.
"The test system, described in more detail in [13], consisted of a hardware test bench connected to a LabVIEW-based software and calculation algorithm." (lines 209-210). In the reviewer's opinion, the article would be better and more interesting if it (even briefly) described the test environment, and not only referred readers to the authors' earlier publication.
Authors response: There is always a balance between including and omitting details from previously published results. A detailed description of the calibration system was considered to be too comprehensive and was therefore only referenced in this work. The focus of this paper has been put on the development and verification of the compensation algorithm. However, additional information and an additional figure regarding the calibration system has been added to chapter 4 - Experimental.
"The implementation of second dimension leads to negligible increasing of memory and computation requirements" (lines 333-334). This claim, in the light of the published results, is unjustified - it is not based on either experimental research or any attempts at estimation. Since this is a relatively important issue and the authors have rightly drawn attention to it, they should develop this topic in their article. The more so that the article for Sensors should be at least 16 pages long, so the authors should add 3 pages anyway to meet the editorial requirements.
Authors response: Thank your for the suggestion about adding additional focus on one of the main advantages with this work. The presented algorithm results in in 4 equations for the compensation factors(K_200,K_500, K_1600 and K_5000), which will in the implementation requires the storage of 8 values. In comparison, by solving the pressure compensation by collection and storage of calibration data in several reference concentrations would quickly result in much higher storage need. E.g. already evaluating 10 reference concentrations at 10 pressures would result in the need of storing 100 values. Additional text has been added in the summary and conclusion which clarify this. This has now been added to the conclusion of the paper. We have seen that there is a general recommendation that the length of articles in Sensors should have a length of 16 pages. Since there exist published papers in Sensors, which are both shorter and longer than this, we have seen this merely as a recommendation and not a requirement. However, the revised paper has been extended by adding additional text and figures.
Round 2
Reviewer 1 Report
The authors have done a lot of work to improve the understanding of the article and the rationale for their motivation. I believe the manuscript can be published in its present form.
Typo - l.59 - Probably a comma [6,4-8] should be put in the citation [64-8].
Author Response
Reviwer 1:
The authors have done a lot of work to improve the understanding of the article and the rationale for their motivation. I believe the manuscript can be published in its present form.
Typo - l.59 - Probably a comma [6,4-8] should be put in the citation [64-8].
It seems that there is issue with figures representation when Track-changes is used:
In the final mode, the representation should be correct according to:
However, the reference list has been updated and this should now read [5-7].

Reviewer 4 Report
Thanks to the authors for correcting the paper. This paper is acceptable, but there are still some shortcomings.
Figure 1 is exactly the same as in the author's previous work ("A practical solution for accurate studies of NDIR gas sensor pressure dependence"). Please redraw Figure 1 or delete it.
Please check the citations of literature throughout the paper, because the numbering in the main text is not always synchronized with the numbering in the References section.
Author Response
Reviewer 4:
Thanks to the authors for correcting the paper. This paper is acceptable, but there are still some shortcomings.
Figure 1 is exactly the same as in the author's previous work ("A practical solution for accurate studies of NDIR gas sensor pressure dependence"). Please redraw Figure 1 or delete it.
Figure1 was, in the revised paper, added based on the request from other reviewers to include more background material rather than only references. However, Figure 1 has based on your new comment been re-drawn slightly. Additional information’s has added in the figure caption.
Please check the citations of literature throughout the paper, because the numbering in the main text is not always synchronized with the numbering in the References section
All citation numbering has been revised and corrected.